# Diagnostic pathways for breast cancer in 10 International Cancer Benchmarking Partnership (ICBP) jurisdictions: an international comparative cohort study based on questionnaire and registry data

Peter Vedsted,[1,2] David Weller,[3] Alina Zalounina Falborg,[2] Henry Jensen ![ORCID],[2] Jatinderpal Kalsi,[4] David Brewster,[5] Yulan Lin,[6] Anna Gavin,[7] Andriana Barisic,[8] Eva Grunfeld,[9] Mats Lambe,[10] Martin Malmberg,[11] Donna Turner,[12] Elizabeth Harland,[13] Breann Hawryluk,[14] Rebecca-Jane Law ![ORCID],[15] Richard D Neal ![ORCID],[16] Victoria White,[17,18] Rebecca Bergin,[19] Samantha Harrison,[20] Usha Menon ![ORCID],[21] The ICBP Module 4 Working Group,[20] ICBP Module 4 Academic Reference Group

PV and DW contributed equally.

For numbered affiliations see end of article.

**Correspondence to**
Professor Peter Vedsted;
p.vedsted@clin.au.dk

## ABSTRACT

**Objectives** A growing body of evidence suggests longer time between symptom onset and start of treatment affects breast cancer prognosis. To explore this association, the International Cancer Benchmarking Partnership Module 4 examined differences in breast cancer diagnostic pathways in 10 jurisdictions across Australia, Canada, Denmark, Norway, Sweden and the UK.

**Setting** Primary care in 10 jurisdictions.

**Participant** Data were collated from 3471 women aged >40 diagnosed for the first time with breast cancer and surveyed between 2013 and 2015. Data were supplemented by feedback from their primary care physicians (PCPs), cancer treatment specialists and available registry data.

**Primary and secondary outcome measures** Patient, primary care, diagnostic and treatment intervals.

**Results** Overall, 56% of women reported symptoms to primary care, with 66% first noticing lumps or breast changes. PCPs reported 77% presented with symptoms, of whom 81% were urgently referred with suspicion of cancer (ranging from 62% to 92%; Norway and Victoria). Ranges for median patient, primary care and diagnostic intervals (days) for symptomatic patients were 3–29 (Denmark and Sweden), 0–20 (seven jurisdictions and Ontario) and 8–29 (Denmark and Wales). Ranges for median treatment and total intervals (days) for all patients were 15–39 (Norway, Victoria and Manitoba) and 4–78 days (Sweden, Victoria and Ontario). The 10% longest waits ranged between 101 and 209 days (Sweden and Ontario).

**Conclusions** Large international differences in breast cancer diagnostic pathways exist, suggesting some jurisdictions develop more effective strategies to optimise pathways and reduce time intervals. Targeted awareness interventions could also facilitate more timely diagnosis of breast cancer.

## STRENGTHS AND LIMITATIONS OF THIS STUDY

⇒ The study used an internationally standardised survey methodology to explore and compare key intervals from symptom onset to start of treatment.
⇒ Comprehensive data were extracted from state/provincial-level cancer registries and other sources to create as complete a record as possible of patient pathways to diagnosis and treatment.
⇒ Minimal recall bias was achieved through triangulation of different data sources and by ensuring that the patients received the questionnaire with a limited time window after the cancer diagnosis.
⇒ Some jurisdictions were not able to recruit a sufficient number of patients to meet power requirements.
⇒ The cohort is not representative of all patients with breast cancer as there was high self-selection of patients with early-stage breast cancer.

## INTRODUCTION

Breast cancer is the most common cancer among women in Western countries. The incidence of female breast cancer is around 80 per 100 000 (standardised world population); by the age of 80, approximately 12% of women will have received a breast cancer diagnosis.[1] Although the prognosis of breast cancer has improved dramatically in recent decades,[2] there are still important differences in disease-specific mortality and survival, both between and within countries.[3 4] For example, in 2010–2014, the 5-year net survival varied from 85.6% in the UK to 88.8% in Sweden and 89.5% in Australia.[2]

⇒ Screening.
⇒ Symptomatic.
  ⇒ Visit PCP.
  ⇒ Visit PCP and then A&E department.
  ⇒ Direct to A&E.
  ⇒ Investigation for another problem.
⇒ Other/unknown routes to diagnosis.

A&E, accident and emergency; PCP, primary care physician.

Differences in survival may relate to timely cancer diagnosis and access to optimal treatment and can also affect patient experience and healthcare costs.[5–10] Examining international differences in routes to diagnosis and treatment, together with time intervals from first noticing symptoms until start of treatment, may help explain these differences.

Women differ in their help-seeking behaviours for breast cancer symptoms.[11] These behaviours might influence the time interval between first noticing symptoms until they present to a healthcare professional.[12] Further, the organisational features of a healthcare system might affect the way patients seek help, how healthcare professionals respond to these symptoms, and when and how they can refer for further investigations.[13–16] In some countries, breast cancer diagnosis is standardised or expedited with urgent referrals, where significant breast cancer-specific 'red flag symptoms' are present; further, many countries have now implemented breast cancer screening programmes.[17] Therefore, routes to diagnosis and time intervals from first noticing symptoms to breast cancer diagnosis and treatment may vary between countries. This has consequences on patient outcomes and experience, as well as healthcare costs.

To date almost all studies on breast cancer routes to diagnosis have been based within a single country[18]; international comparisons can shed light on factors which underpin the observed differences. Accordingly, in this paper we explore pathways to diagnosis and treatment

**Box 2   Time intervals**

⇒ First onset of symptoms: the timepoint when first bodily changes and/or symptoms were first noticed by the patient.
⇒ First presentation to healthcare: the timepoint at which the PCP noted a symptom that in retrospect could be due to the underlying cancer.
⇒ First referral to secondary care: the timepoint at which the PCP referred the patient (and responsibility) to secondary/specialist care.
⇒ Date of diagnosis, following the International Agency for Research on Cancer definition.[21]
⇒ Date of treatment start: when the patient started curative/palliative treatment (in Manitoba, only curative treatment was registered).

PCP, primary care physician.

for female breast cancer in 10 jurisdictions across six countries.

## METHODS
We undertook an international comparative cohort study based on questionnaire and registry data on female patients aged 40 or over with first-time, newly diagnosed breast cancer.

### Study context and management
Within the International Cancer Benchmarking Partnership (ICBP), Module 4 aims to explore differences in routes to diagnosis for four cancers; the methods used are described in detail elsewhere.[19 20] Briefly, the study was undertaken in 10 jurisdictions across six countries: Australia (Victoria), Canada (Manitoba and Ontario), Denmark, Norway, Sweden and the UK (England, Northern Ireland, Scotland and Wales). We recruited jurisdictions with universal healthcare access (with coverage including breast cancer investigations and screening programmes) and showing variation in breast cancer survival.[20]

### Identification of study population
Eligible patients were women aged 40 years or more with a first diagnosis of invasive breast cancer (International Classification of Diseases 10th Revision (ICD-10) codes C50.0—C50.9), irrespective of symptomatic or screen-based diagnosis. Women with previous diagnoses of other cancers were included, but those with a previous diagnosis of breast cancer or synchronous breast cancer were excluded.

Participants were identified via cancer registries and hospital databases, although processes varied locally. Processes were tested in each jurisdiction to ensure adaptation of data collection, questionnaire logistics and data management to the local settings.[20] Each jurisdiction aimed to recruit 200 symptomatic patients with breast cancer, irrespective of the number of screen-based diagnoses. In Ontario and Victoria, the sampling continued beyond 200 patients for use in local studies. In Northern Ireland, the majority of screen-detected women were excluded at the sampling stage by the cancer registry.

### Patient and public involvement
The development of the research question, the questionnaire and the presentation was done in collaboration with patient representatives and cancer charities. Both in the jurisdictions where it was developed, the questionnaire was evaluated and validated among patients and general practitioners and as part of the collaborative work in the ICBP. Patients were involved in the design of how to approach a person with a new cancer diagnosis. The results of this study will be disseminated through the various cancer charities in each jurisdiction.

### Data sources
Data were collected primarily from three questionnaires sent to eligible women (online supplemental additional

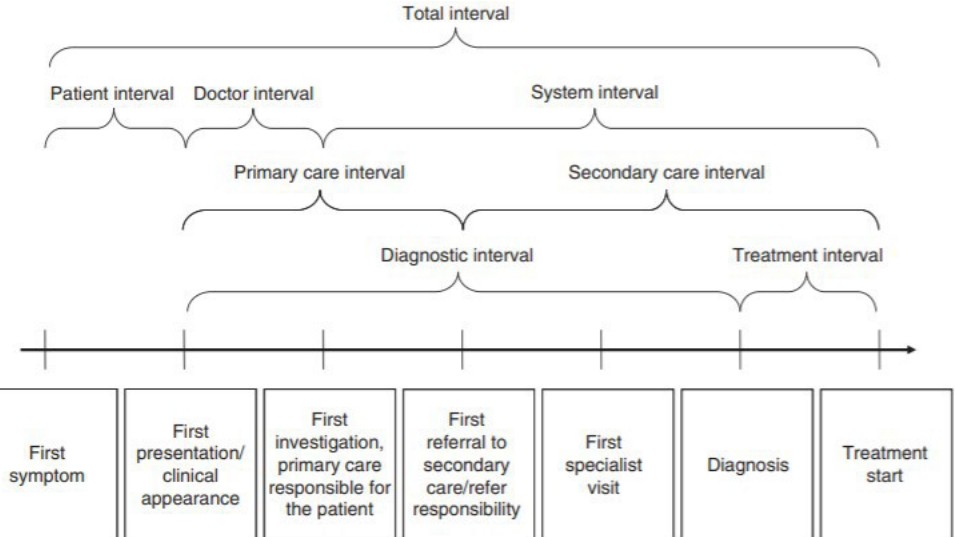

**Figure 1** Diagnostic and treatment intervals.[23]

file 1), their primary care physician (PCP; Online supplemental additional file 2) and their cancer treatment specialists (CTS; Online supplemental additional file 3). This was supplemented with registry data (including diagnosis date, tumour stage and screening) in all jurisdictions except Ontario and Norway.[19] Data were anonymised prior to transfer to Aarhus University (Denmark) for analysis.

### Questionnaire and registry data
The validated questionnaires included items on routes to diagnosis, symptoms, specific milestones from first noticing symptoms to starting treatment, and patient sociodemographics and comorbidity.[20] Eligible patients were identified from cancer registries, and patients were either contacted directly or via their PCP.

Women were sent questionnaires 3–9 months after diagnosis; PCPs and CTS were sent questionnaires after patients consented to participate in the study. Completed questionnaires were returned to the local research teams. Where possible, registries and disease databases provided additional checks and information on date of diagnosis, screening status and start of treatment. Date of diagnosis was based on the International Agency for Research on Cancer hierarchy, and tumour stage using the tumour, node, metastasis classification.[21 22]

### Data handling
Curation of questionnaire data was undertaken primarily at a single location (Aarhus University, Denmark) to ensure uniform application across jurisdictions.

A set of algorithms (or 'rules') to standardise the validation and combination of data from the different sources was developed (online supplemental additional file 4); they employed a hierarchy principle with regard to the order in which different data sources were used. Any queries were discussed and resolved with the study team.

### Routes to diagnosis
Differences in the diagnostic routes were investigated drawing on the checklist from the Aarhus statement.[23] Routes to diagnosis are outlined in Box 1.

### Measures of time intervals
Time interval definitions were adapted from the Aarhus statement and included the timepoints outlined in Box 2[23]; the intervals are represented graphically in figure 1.

For screen-detected women, we used the date of the screening test as the start of the diagnostic and total interval measurements. The analyses included intervals for both symptomatic patients with screen-detected breast cancer and all patients with breast cancer. We included a categorical variable where the symptomatic women reported the waiting time for their appointment with the PCP. Cancers were only considered to be screen-detected if the woman or the PCP had indicated a screening route and there were no symptoms nor symptomatic pathway reported. For Wales, England, Scotland, Northern Ireland, Denmark, Manitoba and Sweden, the distinction between a screen-detected and a non-screen-detected breast cancer was based on registry data from public screening programmes (table 1).

### Covariates
General health status was measured using the self-reported general health item from the 12-Item Short Form Health Survey.[24] Comorbidity was assessed through the patient questionnaire as presence of any of four chronic diseases (heart or lung diseases, stroke, or diabetes) and categorised into 'none', 'medium' (one or two) or 'high' (three or four). Educational level (from the patient questionnaire) was categorised as 'low' (vocational school or lower) and 'high' (university or higher). Smoking was categorised into 'current', 'past' or 'never' smoker. Symptoms reported by the PCP were divided into

**Table 1** Characteristics of 3471 women (aged 40+) with a first diagnosis of breast cancer (% if nothing else stated)

| | Wales (n=232) | England (n=363) | Scotland (n=371) | Northern Ireland (n=331) | Denmark (n=369) | Ontario (n=403) | Norway (n=372) | Manitoba (n=368) | Victoria (n=370) | Sweden (n=292) | Total (N=3471) |
|---|---|---|---|---|---|---|---|---|---|---|---|
| Age (years), median (IQI) | 63 (53–70) | 63 (53–70) | 63 (54–70) | 60 (51–70) | 64 (55–71) | 63 (55–69) | 61 (52–69) | 63 (55–72) | 58 (50–67) | 63 (54–71) | 62 (53–70) |
| Age (years) (symptomatic women), median (IQI) | (n=115) 62 (49–76) | (n=207) 62 (50–75) | (n=208) 62 (52–75) | (n=242) 60 (49–72) | (n=231) 68 (54–77) | (n=201) 61 (53–69) | (n=235) 61 (50–73) | (n=205) 62 (50–75) | (n=198) 56 (48–68) | (n=120) 68 (54–77) | (n=1962) 62 (50–74) |
| Age (years) (screen-detected women), median (IQI) | (n=116) 63 (58–68) | (n=155) 64 (58–68) | (n=162) 63 (58–67) | (n=86) 60 (54–67) | (n=134) 62 (56–66) | (n=198) 65 (59–69) | (n=132) 62 (57–67) | (n=162) 65 (59–70) | (n=172) 61 (53–67) | (n=171) 61 (54–69) | (n=1488) 63 (57–68) |
| Health state | | | | | | | | | | | |
| Good | 81.0 | 81.0 | 79.3 | 77.7 | 78.1 | 90.3 | 74.7 | 85.6 | 88.4 | 80.8 | 81.8 |
| Fair | 14.7 | 13.0 | 13.7 | 14.2 | 14.6 | 8.2 | 17.5 | 9.2 | 9.5 | 12.3 | 12.6 |
| Poor | 3.5 | 4.7 | 5.1 | 5.4 | 3.5 | 1 | 6.2 | 2.7 | 1.6 | 3.8 | 3.7 |
| Missing | 0.9 | 1.3 | 1.9 | 2.7 | 3.8 | 0.5 | 1.6 | 2.5 | 0.5 | 3.1 | 1.9 |
| Comorbidity* | | | | | | | | | | | |
| No | 74.1 | 75.1 | 69.0 | 78.6 | 62.9 | 61.3 | 73.9 | 70.4 | 76.5 | 80.5 | 71.8 |
| Medium | 24.6 | 23.1 | 29.4 | 19.9 | 34.1 | 25.1 | 23.7 | 27.7 | 23.2 | 16.1 | 25.0 |
| High | 0.9 | 0.8 | 1.1 | 0.9 | 0.8 | 0.0 | 1.1 | 0.5 | 0.0 | 0.3 | 0.6 |
| Missing | 0.4 | 0.8 | 0.5 | 0.6 | 2.2 | 13.6 | 1.3 | 1.4 | 0.3 | 3.1 | 2.6 |
| Education | | | | | | | | | | | |
| Low | 71.6 | 78.2 | 69.0 | 77.0 | 71.8 | 64.3 | 63.7 | 74.7 | 70.0 | 68.8 | 70.8 |
| High | 22.4 | 14.9 | 19.7 | 16.3 | 14.6 | 33.5 | 31.5 | 19.8 | 28.9 | 28.8 | 23.1 |
| Missing | 6.0 | 6.9 | 11.3 | 6.7 | 13.6 | 2.2 | 4.8 | 5.4 | 1.1 | 2.4 | 6.1 |
| Ethnicity | | | | | | | | | | | |
| White | 97.9 | 97.8 | 98.9 | 99.1 | 94.3 | 89.9 | 98.7 | 85.6 | 97.3 | 96.2 | 95.4 |
| Asian | 0.4 | 1.1 | 0.0 | 0.0 | 1.1 | 5.7 | 0.8 | 7.6 | 1.9 | 1.0 | 2.1 |
| Black | 0.4 | 0.8 | 0.3 | 0.0 | 0.0 | 2.0 | 0.0 | 0.5 | 0.0 | 0.0 | 0.4 |
| Other | 0.0 | 0.0 | 0.0 | 0.0 | 0.0 | 1.2 | 0.0 | 6.0 | 0.0 | 0.0 | 0.8 |
| Missing | 1.3 | 0.3 | 0.8 | 0.9 | 4.6 | 1.2 | 0.5 | 0.3 | 0.8 | 2.8 | 1.3 |
| Smoking | | | | | | | | | | | |
| Currently | 7.3 | 7.2 | 8.4 | 11.8 | 15.2 | 4.0 | 15.1 | 7.9 | 5.7 | 19.6 | 9.2 |
| In the past | 35.3 | 38.0 | 41.2 | 33.2 | 41.7 | 43.9 | 40.9 | 44.6 | 36.7 | 41.1 | 39.9 |
| Never | 56.5 | 54.5 | 49.3 | 53.5 | 40.9 | 51.6 | 43.5 | 46.2 | 57.3 | 46.9 | 49.8 |
| Missing | 0.9 | 0.3 | 1.1 | 1.5 | 2.2 | 0.5 | 0.5 | 1.3 | 0.3 | 2.4 | 1.1 |
| Tumour stage (TNM) (all women) | | | | | | | | | | | |
| 0 | 1.7 | 0.6 | 0.0 | 0.0 | 0.3 | 0.3 | 0.0 | 0.0 | 0.3 | 0.0 | 0.3 |

Continued

**Table 1** Continued

| | Wales (n=232) | England (n=363) | Scotland (n=371) | Northern Ireland (n=331) | Denmark (n=369) | Ontario (n=403) | Norway (n=372) | Manitoba (n=368) | Victoria (n=370) | Sweden (n=292) | Total (N=3471) |
|---|---|---|---|---|---|---|---|---|---|---|---|
| I | 50.4 | 44.9 | 47.4 | 45.3 | 45.0 | 51.6 | 11.3 | 50.8 | 52.7 | 62.0 | 45.7 |
| II | 37.9 | 36.4 | 35.6 | 35.1 | 33.1 | 38.0 | 4.6 | 34.5 | 34.6 | 31.5 | 31.9 |
| III | 5.6 | 8.3 | 7.8 | 13.6 | 9.5 | 8.2 | 1.9 | 10.3 | 10.8 | 4.1 | 8.1 |
| IV | 0.9 | 3.0 | 4.6 | 2.7 | 2.4 | 1.0 | 0.8 | 1.6 | 1.1 | 0.7 | 1.9 |
| Missing | 3.5 | 6.9 | 4.5 | 3.3 | 9.8 | 1.0 | 81.5 | 2.7 | 0.5 | 1.7 | 12.1 |
| Tumour stage (TNM) (symptomatic women) | (n=115) | (n=207) | (n=208) | (n=242) | (n=231) | (n=201) | (n=235) | (n=205) | (n=198) | (n=120) | (n=1966) |
| 0 | 1.7 | 1.0 | 0.0 | 0.0 | 0.0 | 0.0 | 0.0 | 0.0 | 0.5 | 0.0 | 0.3 |
| I | 33.0 | 34.8 | 32.7 | 37.2 | 36.4 | 32.8 | 8.1 | 32.7 | 37.9 | 40.8 | 32.1 |
| II | 49.6 | 41.6 | 45.7 | 39.3 | 35.1 | 50.0 | 5.5 | 48.3 | 44.4 | 50.8 | 39.5 |
| III | 8.7 | 10.6 | 9.6 | 17.4 | 11.7 | 14.0 | 2.6 | 13.2 | 15.2 | 3.3 | 11.0 |
| IV | 1.7 | 4.4 | 6.7 | 2.9 | 3.9 | 2.0 | 0.9 | 2.4 | 1.0 | 1.7 | 2.9 |
| Missing | 5.2 | 7.7 | 5.3 | 3.3 | 13.0 | 0.5 | 83.0 | 3.4 | 1.0 | 3.3 | 14.3 |
| Tumour stage (TNM) (screen-detected women) | (n=116) | (n=155) | (n=162) | (n=86) | (n=134) | (n=198) | (n=132) | (n=162) | (n=172) | (n=171) | (n=1484) |
| 0 | 1.7 | 0.0 | 0.0 | 0.0 | 0.8 | 0.5 | 0.0 | 0.0 | 0.0 | 0.0 | 0.3 |
| I | 68.1 | 58.1 | 66.1 | 67.4 | 59.7 | 69.2 | 16.7 | 74.1 | 69.8 | 77.2 | 63.5 |
| II | 25.9 | 29.7 | 22.8 | 23.3 | 29.1 | 26.3 | 2.3 | 17.3 | 23.3 | 18.1 | 21.9 |
| III | 2.6 | 5.2 | 5.6 | 3.5 | 6.0 | 2.5 | 0.8 | 6.8 | 5.8 | 4.1 | 4.4 |
| IV | 0.0 | 1.3 | 1.9 | 2.3 | 0.0 | 0.0 | 0.0 | 0.6 | 1.2 | 0.0 | 0.7 |
| Missing | 1.7 | 5.8 | 3.7 | 3.5 | 4.5 | 1.5 | 80.3 | 1.2 | 0.0 | 0.6 | 9.3 |

Norwegian data on tumour stage were incomplete, resulting in a high number of missing data on stage.
*Comorbidity coded as none: none reported; medium: 1–2 reported; and high: 3+ reported.
IQI, interquartile interval; TNM, tumour, node, metastasis.

**Table 2** Proportion of screen-detected or symptomatic patients (%) and place of initial symptomatic presentation across jurisdictions

| | Wales (n=232) | England (n=363) | Scotland (n=371) | Northern Ireland* (n=331) | Denmark (n=369) | Ontario (n=403) | Norway (n=373) | Manitoba (n=368) | Victoria (n=370) | Sweden (n=292) | Total (N=3471) |
|---|---|---|---|---|---|---|---|---|---|---|---|
| Symptomatic | 49.6 | 57.0 | 56.1 | 73.1 | 62.6 | 49.9 | 63.2 | 55.7 | 53.5 | 41.1 | 56.5 |
| Visit PCP, visit PCP and then A&E† | 91.3 | 93.7 | 92.3 | 88.0 | 84.4 | 69.2 | 81.3 | 88.3 | 90.9 | 75.0 | 85.6 |
| A&E directly† | 1.7 | 1.0 | 0.5 | 0.4 | 1.3 | 2.5 | 0.9 | 1.0 | 0.5 | 0.8 | 1.0 |
| Investigation for another problem† | 0.9 | 1.0 | 4.8 | 1.2 | 6.1 | 7.5 | 4.7 | 2.4 | 1.5 | 10.0 | 3.9 |
| Other† | 6.1 | 4.3 | 2.4 | 10.3 | 8.2 | 20.9 | 13.2 | 8.3 | 7.1 | 14.2 | 9.5 |
| Screening | 50.0 | 42.7 | 43.7 | 26.0 | 36.3 | 49.1 | 35.5 | 44.0 | 46.5 | 58.6 | 42.9 |
| Other | 0.4 | 0.3 | 0.3 | 0.9 | 1.1 | 1.0 | 1.2 | 0.3 | 0.0 | 0.3 | 0.6 |

*In Northern Ireland, the proportion of screen-detected breast cancers was lower as many of these women were excluded from the eligible group at the start of inclusion.
†Percentage of the symptomatic route.
A&E, accident and emergency; PCP, primary care physician.

'breast cancer specific symptoms' or 'other symptoms'. This was based on symptom coding, drawing on local clinical guidelines.[17 24 25]

### Statistical analysis

Patients with missing data on age, date of diagnosis and/or date of consent were excluded. Quantile regression[26] was used to estimate the differences in intervals at the 50th, 75th and 90th percentiles between all jurisdictions, with Wales as the reference. Intervals were derived by counting days, using the 'qcount' procedure.[27 28] The differences in intervals were calculated as marginal effects after quantile regression by setting the continuous covariates to their mean values and the categorical covariates to their modes.

Due to zero inflation, quantile regression could not be used for the primary care interval. Here a generalised linear model with Poisson family, log link and robust error variance was used to calculate the association based on prevalence ratio, between longer primary care intervals and jurisdiction. Longer intervals were defined as those over 5 or 14 days to test both intervals. The analyses were adjusted for age (as a continuous variable) and comorbidity (as a categorical variable). The significance level was set to 0.05 or less, and 95% CIs were calculated when appropriate. Statistical analyses were carried out using STATA V.14 software.

### Sensitivity and validity analyses

Analyses were repeated using the per-protocol definition of a maximum of 6 months between diagnosis and patient consent, to assess the impact of including patients who consented at 9 months postdiagnosis. To estimate the effect of using patient-reported intervals only, a sensitivity analysis based solely on patient questionnaire data was performed.

Kappa coefficients were used to assess the agreement on routes to diagnosis (screening and symptomatic presentation) between the different data sources. Lin's concordance correlation coefficient (CCC) was used to assess the agreement on dates between the different data sources.[29]

### RESULTS

### Patient characteristics and participation

Across all jurisdictions, 15 421 eligible women were identified between May 2013 and August 2015. A total of 4593 (39.9%) of those contacted returned completed questionnaires, of which 3471 (75.6%) were included in the analyses. The patient flow, with identification, exclusion and responses for each jurisdiction, is detailed in online supplemental additional file 4. The response rates ranged from 24.1% in Norway to 77.9% in Denmark (online supplemental table 1). Participating women were younger and had a less advanced tumour stage distribution than the total sample of women eligible for inclusion (online

**Table 3** The first symptoms reported by women* and the first presenting symptoms reported by PCPs (%)

| | Wales | England | Scotland | Northern Ireland* | Denmark | Ontario | Norway | Manitoba | Victoria | Sweden† | Total |
|---|---|---|---|---|---|---|---|---|---|---|---|
| First symptom (reported by women) | (n=115) | (n=207) | (n=208) | (n=242) | (n=231) | (n=201) | (n=235) | (n=205) | (n=198) | (n=120) | (N=1962) |
| Lump/swelling/thickening of breast | 76 | 70 | 75 | 62 | 63 | 63 | 55 | 60 | 71 | 73 | 66 |
| Change in the nipple | 9 | 15 | 11 | 14 | 11 | 10 | 13 | 6 | 6 | 14 | 11 |
| Fatigue | 16 | 10 | 9 | 12 | 5 | 17 | 11 | 12 | 14 | 9 | 11 |
| Change in the size and contour | 12 | 13 | 10 | 10 | 5 | 16 | 8 | 10 | 15 | 11 | 11 |
| Pain/tenderness in the breast | 6 | 5 | 4 | 14 | 3 | 5 | 7 | 9 | 15 | 6 | 8 |
| Flattening/indentation in the breast | 5 | 5 | 0.5 | 4 | 4 | 0 | 2 | 2 | 0 | 0.8 | 2 |
| Other | 21 | 23 | 20 | 17 | 10 | 13 | 18 | 19 | 16 | 12 | 17 |
| No symptoms | 8 | 2 | 4 | 8 | 11 | 10 | 9 | 17 | 5 | 10 | 8 |
| Missing | 0 | 0.4 | 4 | 6 | 10 | 2 | 13 | 4 | 1 | 0 | 5 |
| Presenting symptom (reported by PCP) | (n=97) | (n=169) | (n=160) | (n=0) | (n=153) | (n=74) | (n=109) | (n=123) | (n=128) | (n=0) | (n=1013) |
| Lump/swelling/thickening of breast | 73 | 64 | 64 | n/a | 67 | 34 | 55 | 46 | 64 | n/a | 60 |
| Change in the nipple | 2 | 8 | 5 | n/a | 5 | 7 | 5 | 0.8 | 6 | n/a | 5 |
| Fatigue | 0 | 0 | 0 | n/a | 0 | 0 | 0.9 | 0.8 | 0 | n/a | 0.2 |
| Change in the size and contour | 4 | 4 | 0 | n/a | 6 | 0 | 0.9 | 3 | 5 | n/a | 3 |
| Pain/tenderness in the breast | 10 | 6 | 4 | n/a | 7 | 7 | 4 | 10 | 13 | n/a | 7 |
| Flattening/indentation in the breast | 2 | 2 | 1 | n/a | 0 | 3 | 2 | 3 | 0 | n/a | 2 |
| Other | 13 | 12 | 9 | n/a | 7 | 24 | 13 | 13 | 2 | n/a | 11 |
| No symptoms | 1 | 0.6 | 0.6 | n/a | 0.7 | 11 | 0 | 12 | 2 | n/a | 3 |
| Missing | 8 | 11 | 17 | n/a | 18 | 18 | 21 | 24 | 20 | n/a | 17 |
| Cancer specificity of presenting symptom‡ | | | | | | | | | | | |
| Cancer-specific symptom | 88 | 85 | 79 | n/a | 78 | 69 | 70 | 59 | 78 | n/a | 77 |
| Non-specific symptom | 3 | 3 | 4 | n/a | 3 | 3 | 9 | 4 | 0 | n/a | 4 |
| No symptoms/missing | 9 | 12 | 18 | n/a | 18 | 28 | 21 | 37 | 22 | n/a | 20 |

*These figures represent only women who had a symptom-based cancer diagnosis. More than one symptom could be stated (sum over 100%).
†Sweden and Northern Ireland did not include PCP questionnaire data.
‡Based on the PCP reported symptom/sign and categorised by two researchers.
n/a, not applicable; PCP, primary care physician.

**Table 4** Time intervals (days) for each jurisdiction depicted as median, 75th and 90th percentiles

| | | Wales | England | Scotland | Northern Ireland | Denmark | Ontario | Norway | Manitoba | Victoria | Sweden* |
|---|---|---|---|---|---|---|---|---|---|---|---|
| Patient interval (symptomatic) | n | 111 | 201 | 198 | 219 | 213 | 170 | 206 | 172 | 189 | 101 |
| | Median | 11 | 7 | 7 | 8 | 3 | 19 | 12 | 14 | 7 | 29 |
| | 75th percentile | 34 | 30 | 30 | 31 | 22 | 58 | 48 | 47 | 31 | 56 |
| | 90th percentile | 73 | 92 | 88 | 114 | 63 | 142 | 157 | 86 | 117 | 90 |
| Primary care interval (symptomatic) | n | 97 | 167 | 160 | 172 | 141 | 70 | 99 | 109 | 142 | n/a |
| | Median | 0 | 0 | 0 | 0 | 0 | 20 | 0 | 17 | 7 | |
| | 75th percentile | 0 | 0 | 1 | 0 | 0 | 37 | 0 | 30 | 15 | |
| | 90th percentile | 3 | 7 | 6 | 3 | 10 | 75 | 14 | 82 | 38 | |
| Diagnostic interval (symptomatic) | n | 111 | 200 | 197 | 216 | 207 | 166 | 203 | 184 | 191 | 101 |
| | Median | 29 | 12 | 19 | 14 | 8 | 25 | 20 | 28 | 13 | 13 |
| | 75th percentile | 54 | 18 | 35 | 21 | 26 | 56 | 37 | 42 | 21 | 24 |
| | 90th percentile | 92 | 36 | 49 | 49 | 49 | 202 | 71 | 79 | 46 | 48 |
| Diagnostic interval (screen-detected) | n | 113 | 153 | 159 | 86 | 131 | 184 | 126 | 125 | 152 | 158 |
| | Median | 25 | 15 | 20 | 19 | 24 | 26 | 27 | 23 | 21 | 15 |
| | 75th percentile | 45 | 21 | 29 | 26 | 36 | 42 | 44 | 43 | 32 | 25 |
| | 90th percentile | 58 | 34 | 42 | 41 | 48 | 66 | 66 | 113 | 40 | 41 |
| Diagnostic interval (all) | n | 224 | 353 | 356 | 302 | 338 | 350 | 329 | 309 | 343 | 259 |
| | Median | 29 | 13 | 20 | 15 | 14 | 26 | 22 | 26 | 16 | 14 |
| | 75th percentile | 49 | 21 | 33 | 21 | 31 | 46 | 39 | 42 | 28 | 25 |
| | 90th percentile | 70 | 36 | 48 | 43 | 49 | 90 | 68 | 94 | 45 | 41 |
| Treatment interval (symptomatic) | n | 115 | 203 | 202 | 241 | 230 | 201 | 216 | 198 | 196 | 113 |
| | Median | 24 | 29 | 24 | 21 | 20 | 35 | 19 | 38 | 14 | 22 |
| | 75th percentile | 33 | 41 | 37 | 29 | 29 | 49 | 27 | 55 | 24 | 29 |
| | 90th percentile | 45 | 55 | 54 | 40 | 44 | 65 | 34 | 72 | 33 | 42 |
| Treatment interval (screen-detected) | n | 115 | 155 | 158 | 86 | 134 | 195 | 120 | 155 | 170 | 165 |
| | Median | 27 | 31 | 34 | 24 | 19 | 35 | 13 | 39 | 19 | 21 |
| | 75th percentile | 36 | 40 | 44 | 35 | 29 | 48 | 21 | 51 | 28 | 29 |
| | 90th percentile | 49 | 62 | 69 | 44 | 36 | 65 | 28 | 66 | 38 | 40 |
| Treatment interval (all) | n | 231 | 359 | 361 | 330 | 368 | 400 | 341 | 353 | 366 | 278 |
| | Median | 25 | 30 | 29 | 22 | 20 | 35 | 15 | 39 | 15 | 22 |
| | 75th percentile | 35 | 41 | 41 | 31 | 29 | 48 | 24 | 54 | 27 | 29 |
| | 90th percentile | 46 | 57 | 61 | 41 | 41 | 65 | 33 | 71 | 36 | 41 |

Continued

**Table 4** Continued

| | | Wales | England | Scotland | Northern Ireland | Denmark | Ontario | Norway | Manitoba | Victoria | Sweden* |
|---|---|---|---|---|---|---|---|---|---|---|---|
| Total interval (symptomatic) | n | 104 | 190 | 186 | 205 | 189 | 173 | 184 | 155 | 177 | 98 |
| | Median | 70 | 57 | 58 | 50 | 42 | 92 | 54 | 92 | 42 | 71 |
| | 75th percentile | 96 | 82 | 99 | 78 | 73 | 158 | 121 | 128 | 89 | 101 |
| | 90th percentile | 218 | 138 | 149 | 147 | 170 | 273 | 231 | 188 | 170 | 169 |
| Total interval (all) | n | 217 | 343 | 341 | 291 | 320 | 354 | 298 | 277 | 327 | 253 |
| | Median | 60 | 52 | 55 | 46 | 44 | 78 | 48 | 76 | 42 | 42 |
| | 75th percentile | 81 | 70 | 84 | 69 | 68 | 116 | 79 | 116 | 63 | 68 |
| | 90th percentile | 123 | 114 | 129 | 127 | 118 | 209 | 168 | 182 | 120 | 101 |

*In Sweden, data on the primary care interval were not collected.
n/a, not applicable.

supplemental additional file 5). The characteristics of the included women are detailed in table 1.

## First presentation and symptoms

Table 2 shows the routes to diagnosis; 42.9% had a screen-detected breast cancer, with the highest proportion in Sweden (58.6%) (Note that the proportion of screen-detected cancers in Northern Ireland is lower due to the initial exclusion of screen-detected cancers.). For symptomatic patients, primary care was the place of first presentation for 85.6% of women, ranging from 69.2% in Ontario to 93.7% in England. According to the PCP, 81.1% of symptomatic women were urgently referred with a suspicion of cancer, ranging from 62.4% in Norway to 92.2% in Victoria (data not shown).

Table 3 shows the first symptom or sign reported by women, together with those reported by the PCP (for Sweden, only patient data were collected). Two-thirds of symptomatic women noticed a lump or change in the breast. This was also the most commonly reported sign by PCPs, followed by pain/tenderness and change in the nipple. According to the PCP, 77% of women presented with a symptom or sign indicative of breast cancer. While 1 in 10 women indicated fatigue as a symptom, this was rarely noted by the PCPs.

## Time intervals

The median patient interval for symptomatic women varied from 3 days in Denmark to 29 days in Sweden (see table 4). Based on the quantile regression, patient interval was 7 days shorter in Denmark and 17 days longer in Sweden compared with Wales (figure 2 and online supplemental additional file 6). When comparing the 90th percentile, Norway (157 days) and Ontario (142 days) had the longest patient intervals (table 4). Quantile regression showed that Norway had a significantly longer patient interval for the 90th percentile compared with Wales (74 days) (figure 2).

The median primary care interval for symptomatic women (in days) was 0, except in the two Canadian provinces and in Victoria (table 4). Women in these three jurisdictions had longer primary care intervals (both for >5 and >14 days) than women in Wales; this was statistically significant (table 5).

The median diagnostic interval for symptomatic women ranged from 8 days in Denmark to 29 days in Wales (table 4). At the 90th percentile, the diagnostic interval ranged from 36 days in England to 202 days in Ontario (table 4). Compared with Wales, all jurisdictions except Ontario had shorter diagnostic intervals (figure 2). For women with a screen-detected breast cancer, the time intervals and pattern were similar, although screen-detected patients had significantly shorter 90th percentiles compared with symptomatic patients.

The median treatment intervals for all women ranged from 15 days in Norway and Victoria to 39 days in Manitoba. Women in England, Scotland, Ontario and Manitoba waited more than 28 days (4 weeks) for treatment after

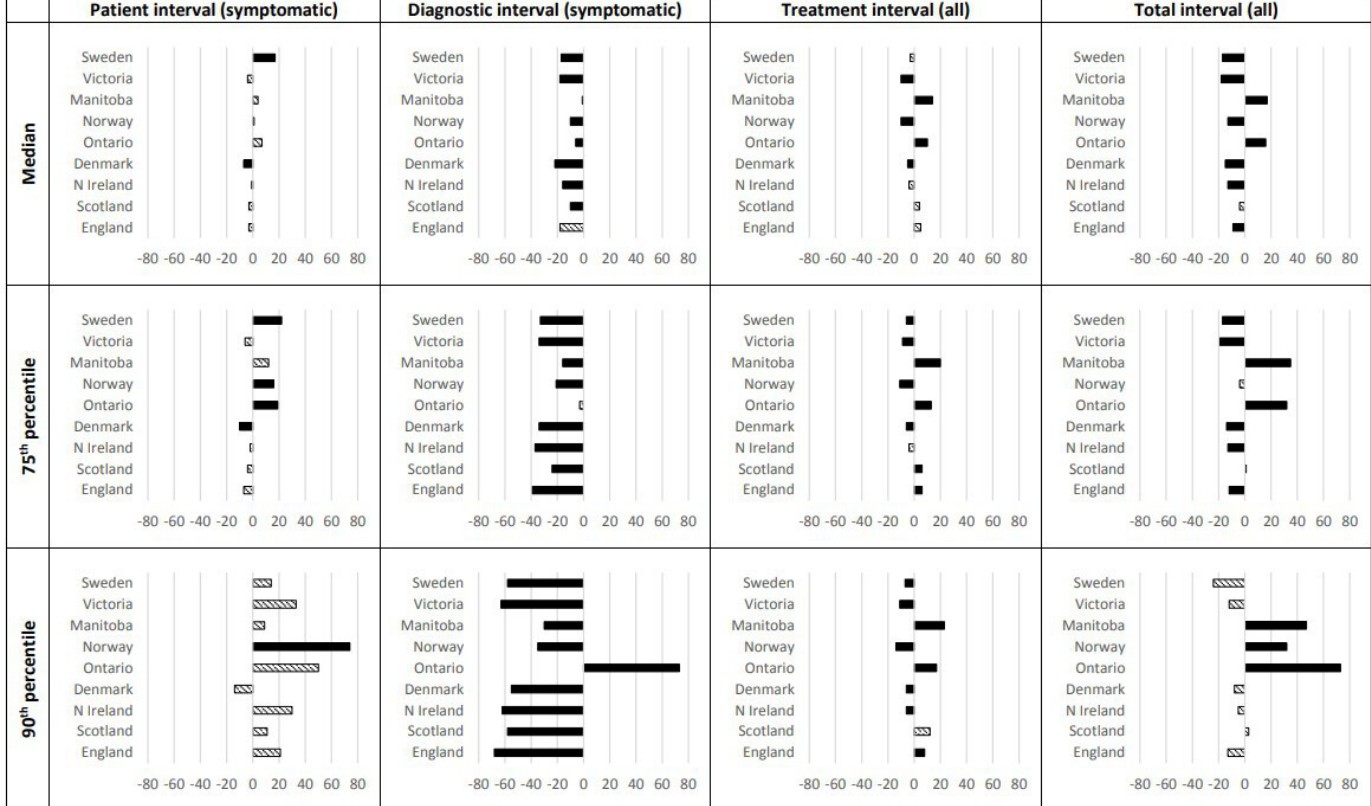

**Figure 2** Time intervals for each jurisdiction compared with Wales (reference). Adjusted for differences in comorbidity and age. Differences in interval lengths (in days) are shown for the median, 75th and 90th percentiles compared with the reference used for the regression analyses, Wales. Wales is represented by the axis, with jurisdictions with shorter intervals shown to the left of the axis and jurisdictions with longer intervals shown to the right of the axis for each graph. Solid-fill bars indicate statistically significant differences compared with Wales. Primary care interval is not shown (see table 6).

diagnosis (table 4). Quantile regression showed smaller statistically significant differences in treatment intervals, with Manitoba and Ontario consistently having longer treatment intervals for the three percentiles (figure 2). The treatment intervals were similar for screen-detected cancers.

The median total interval for all women from first symptom or date of screening test to start of treatment ranged from 42 days in Sweden and Victoria to 78 days in Ontario (table 6). In Manitoba and Ontario, 25% of women had a total interval that was 30 days longer compared with Wales (81 days), whereas 25% of women in England, Northern Ireland, Denmark, Victoria and Sweden had a total interval that was between 12 and 19 days shorter compared with Wales (figure 2 and online supplemental additional file 6). In Ontario, 10% of women waited 73 days or more from first presentation until treatment started compared with women in Wales. Including screen-detected breast cancers made the total interval shorter, with less variation between the jurisdictions.

There were differences in the time taken from women deciding to seek help getting an appointment with a PCP. Getting an appointment 'within one week' was less often reported by women in Sweden (43%), Manitoba (50%), Ontario (58%) and Northern Ireland (64%) compared with women in other jurisdictions (above 70% in the other six

jurisdictions, with 84% in England and Victoria) (data not shown).

## Validity tests

Comparing patient and PCP-reported screening with available screening registry data showed an almost perfect agreement (kappa >0.80) (online supplemental additional file 7). Comparing the dates between the different data sources showed a high agreement between all data sources (patient, PCP, CTS and register where applicable) for all types of dates (CCC=0.94 for date of first presentation to primary care, CCC ≥0.94 for date of diagnosis and CCC=0.93 for date of treatment).

## DISCUSSION
### Main findings

This international survey of patients with breast cancer diagnosed between 2013 and 2015 revealed that 4 out of 10 were screen-detected and more than half of women presented with symptoms, with the majority of these diagnosed after a visit to primary care. Three-quarters of symptomatic patients had a symptom or sign indicative of breast cancer, and roughly 8 in 10 were urgently referred. Thus, despite the existence of screening programmes,

**Table 5** Prevalence ratio (PR)* of experiencing differing primary care intervals for symptomatic patients with breast cancer

| | Wales | England | Scotland | Northern Ireland | Denmark | Ontario | Norway | Manitoba | Victoria | Sweden† |
|---|---|---|---|---|---|---|---|---|---|---|
| | (n=97) | (n=167) | (n=160) | (n=172) | (n=141) | (n=70) | (n=99) | (n=109) | (n=142) | (n=0) |
| **Interval ≥5days** | | | | | | | | | | |
| PR (95% CI) | Ref | 1.7 (0.7 to 4.2) | 2.0 (0.8 to 4.9) | 1.3 (0.5 to 3.3) | 2.0 (0.8 to 4.8) | **12.9** (5.9 to 28.2) | 2.2 (0.9 to 5.6) | **12.9** (5.9 to 28.1) | **11.1** (5.1 to 24.4) | n/a |
| **Interval ≥14 days** | | | | | | | | | | |
| PR (95% CI) | Ref | 1.6 (0.5 to 4.9) | 1.5 (0.5 to 4.7) | 1.4 (0.5 to 4.3) | 2.1 (0.7 to 6.3) | **15.1** (5.7 to 40.1) | 2.4 (0.8 to 7.4) | **14.3** (5.4 to 37.8) | **6.7** (2.5 to 18.1) | n/a |

Bold indicates significance level at p≤0.05.
*Adjusted for differences in age and comorbidity among patients, with Wales as reference.
†In Sweden, no information on primary care interval was collected.
n/a, not applicable; ref, reference.

the majority still had their breast cancer diagnosed based on 'red flag' symptoms.

The median time from women first noticing symptoms to start of treatment varied from 42 to 92 days across jurisdictions; this is attributable mainly to differences after first presentation. Less variation is seen in patient interval across jurisdictions.

We found that in some jurisdictions there is a 'tail' of women with excessively long waiting times from first noticing symptoms to presentation to the healthcare system.

### Strengths and weaknesses

A major strength is our novel use of a standardised questionnaire survey in several countries to systematically examine routes to diagnosis and treatment. To ensure validity across jurisdictions, we drew on existing instruments and went through an extensive process of cognitive testing, piloting, translation and adaptation.[20] Data were enriched with information from national or state/provincial-level cancer registries via our ICBP Module 4 Working Group (online supplemental additional file 8) for screening status, date of diagnosis and tumour stage. We developed algorithms to identify screen-detected cases, place of first presentation and time intervals, which showed good agreement for those jurisdictions where validation was possible. Using registries and undertaking clinical validation of patients with breast cancer ensured minimal selection and information biases. It also made it possible to exclude women who previously had breast cancer, thus providing a reasonably homogeneous group of patients.

We included all patients with a first diagnosis of breast cancer, irrespective of route to diagnosis, and recruited at least 200 symptomatic cases per jurisdiction. We developed and implemented validated rules for identifying the screening route, which showed high agreement with screening registries.

There were different classification systems for ethnicity and education across jurisdictions. We excluded these variables from the regression model to avoid introducing information bias in the model, which would have compromised its validity. Although this likely induced some residual confounding, it is unlikely that educational or ethnic differences could have produced the observed variation in time intervals.

We used a set of rules to ensure validity and consistency and to preserve statistical precision.[20] The validity and sensitivity analyses suggested this approach was effective. To minimise misclassification, data interpretation during data entry by the local teams was reduced to an absolute minimum and all apparent data errors were checked against source data by local teams.

The effect of recall bias was minimised by triangulation of data sources (eg, patient, PCP, CTS and registry data) and by ensuring that the women received the questionnaire within a limited time window after diagnosis.

**Table 6** Differences in interval lengths (days) between Wales, the reference, and the other nine jurisdictions

| | | Wales | England | Scotland | Northern Ireland | Denmark | Ontario | Norway | Manitoba | Victoria | Sweden |
|---|---|---|---|---|---|---|---|---|---|---|---|
| Patient interval (symptomatic) | n | 111 | 201 | 198 | 219 | 213 | 170 | 206 | 172 | 189 | 101 |
| | Median | 11 (ref) | −3 (−9, 3) | −3 (−8, 3) | −1 (−8, 6) | **−7** (−12, −2) | 7 (−1, 16) | 1 (−7, 9) | 4 (−2, 11) | −4 (−9, 2) | **17** (11, 22) |
| | 75th percentile | 34 (ref) | −7 (−21, 7) | −4 (−12, 3) | −2 (−8, 5) | **−10** (−16, −4) | **19** (5, 33) | **16** (7, 25) | 12 (−1, 25) | −6 (−16, 4) | **22** (8, 36) |
| | 90th percentile | 73 (ref) | 21 (−19, 61) | 11 (−45, 66) | 30 (−13, 73) | −14 (−64, 35) | 50 (−9, 110) | **74** (31, 117) | 9 (−35, 54) | 33 (−13, 80) | 14 (−31, 60) |
| Diagnostic interval (symptomatic) | n | 111 | 200 | 197 | 216 | 207 | 166 | 203 | 184 | 191 | 101 |
| | Median | 29 (ref) | **−18** (−22, −15) | **−10** (−14, −6) | **−16** (−19, −13) | **−22** (−25, −18) | **−6** (−9, −2) | **−10** (−15, −5) | −1 (−5, 4) | **−18** (−22, −14) | **−17** (−23, −12) |
| | 75th percentile | 54 (ref) | **−39** (−45, −33) | **−24** (−29, −19) | **−37** (−42, −33) | **−34** (−40, −28) | −3 (−13, 6) | **−21** (−36, −6) | **−16** (−23, −8) | **−34** (−52, −15) | **−33** (−39, −27) |
| | 90th percentile | 92 (ref) | **−68** (−88, −48) | **−58** (−91, −25) | **−62** (−80, −45) | **−55** (−75, −35) | **73** (35, 112) | **−35** (−62, −8) | **−30** (−46, −15) | **−63** (−78, −47) | **−58** (−81, −35) |
| Diagnostic interval (screen-detected) | n | 113 | 153 | 159 | 86 | 131 | 184 | 126 | 125 | 152 | 158 |
| | Median | 25 (ref) | −11 (−22, 1) | −7 (−18, 5) | −7 (−19, 4) | −1 (−17, 15) | 0 (−12, 12) | 1 (−11, 13) | −2 (−15, 10) | −5 (−17, 6) | −12 (−24, 0) |
| | 75th percentile | 45 (ref) | **−24** (−36, −13) | **−17** (−28, −6) | **−20** (−31, −8) | −9 (−21, 2) | −3 (−15, 8) | −1 (−15, 13) | −4 (−15, 7) | **−14** (−25, −3) | **−21** (−34, −7) |
| | 90th percentile | 58 (ref) | **−23** (−30, −16) | **−16** (−22, −11) | **−19** (−26, −12) | **−10** (−14, −6) | 9 (−6, 24) | 8 (−3, 20) | **55** (38, 71) | **−17** (−26, −8) | **−17** (−20, −13) |
| Diagnostic interval (all) | n | 224 | 353 | 356 | 302 | 338 | 350 | 329 | 309 | 343 | 259 |
| | Median | 29 (ref) | **−14** (−16, −12) | **−8** (−11, −6) | **−13** (−15, −11) | **−14** (−16, −11) | **−2** (−5, 1) | **−6** (−9, −2) | −1 (−5, 2) | **−12** (−15, −10) | **−14** (−17, −11) |
| | 75th percentile | 49 (ref) | **−30** (−34, −25) | **−18** (−23, −14) | **−29** (−33, −25) | **−20** (−25, −16) | −5 (−15, 4) | **−11** (−20, −2) | **−8** (−12, −3) | **−22** (−26, −19) | **−25** (−32, −17) |
| | 90th percentile | 70 (ref) | **−35** (−39, −31) | **−23** (−30, −16) | **−26** (−34, −19) | **−21** (−40, −2) | **20** (9, 31) | −1 (−16, 14) | **22** (7, 38) | **−27** (−40, −13) | **−29** (−36, −21) |
| Treatment interval (symptomatic) | n | 115 | 203 | 202 | 241 | 230 | 201 | 216 | 198 | 196 | 113 |
| | Median | 24 (ref) | **6** (1, 11) | 1 (−3, 5) | −3 (−7, 1) | −3 (−7, 1) | **12** (8, 16) | −5 (−11, 1) | **15** (11, 19) | **−10** (−14, −6) | −2 (−6, 2) |
| | 75th percentile | 33 (ref) | **8** (4, 12) | 4 (−1, 9) | −3 (−8, 1) | −4 (−9, 0) | **16** (11, 21) | **−6** (−9, −2) | **22** (18, 26) | **−9** (−16, −2) | −4 (−13, 5) |
| | 90th percentile | 45 (ref) | 10 (−1, 20) | 11 (−4, 26) | −4 (−19, 11) | −1 (−14, 12) | **20** (10, 31) | −11 (−23, 2) | **28** (10, 46) | −11 (−26, 4) | −3 (−15, 10) |
| Treatment interval (screen-detected) | n | 115 | 155 | 158 | 86 | 134 | 195 | 120 | 155 | 170 | 165 |
| | Median | 27 (ref) | **5** (1, 8) | **8** (4, 11) | 0 (−5, 5) | **−6** (−10, −1) | **9** (7, 12) | **−12** (−15, −10) | **12** (9, 16) | −5 (−12, 1) | **−3** (−6, −1) |
| | 75th percentile | 36 (ref) | 4 (−2, 10) | 10 (−3, 22) | 0 (−6, 7) | −7 (−16, 3) | **11** (3, 20) | **−15** (−27, −4) | **15** (7, 23) | −7 (−16, 2) | −6 (−17, 5) |
| | 90th percentile | 49 (ref) | 5 (−8, 18) | **15** (4, 25) | −9 (−20, 3) | **−14** (−26, −2) | 9 (−4, 22) | **−24** (−35, −13) | 12 (−1, 24) | **−14** (−27, −1) | −11 (−21, 0) |
| Treatment interval (all) | n | 231 | 359 | 361 | 330 | 368 | 400 | 341 | 353 | 366 | 278 |
| | Median | 25 (ref) | 5 (0, 11) | 4 (0, 8) | −4 (−8, 1) | **−5** (−9, −1) | **10** (6, 14) | **−10** (−14, −6) | **14** (9, 18) | **−10** (−14, −6) | −3 (−7, 0) |
| | 75th percentile | 35 (ref) | **6** (2, 10) | **6** (1, 10) | −4 (−8, 0) | **−6** (−10, −2) | **13** (10, 17) | **−11** (−16, −6) | **20** (16, 23) | **−9** (−13, −5) | **−6** (−10, −2) |
| | 90th percentile | 46 (ref) | **8** (2, 14) | 12 (−1, 26) | **−6** (−10, −2) | **−6** (−9, −2) | **17** (11, 23) | **−14** (−17, −12) | **23** (16, 29) | **−11** (−14, −9) | **−7** (−10, −4) |
| Total interval (symptomatic) | n | 104 | 190 | 186 | 205 | 189 | 173 | 184 | 155 | 177 | 98 |
| | Median | 70 (ref) | **−14** (−24, −3) | **−12** (−23, −2) | **−20** (−31, −8) | **−25** (−38, −11) | **28** (15, 42) | −15 (−34, 5) | **22** (12, 31) | **−28** (39, −17) | −1 (−11, 10) |
| | 75th percentile | 96 (ref) | **−16** (−27, −5) | −5 (−18, 9) | **−18** (−33, −3) | **−25** (−39, −12) | **69** (31, 107) | **27** (8, 46) | **30** (8, 53) | −13 (−27, 0) | 3 (−30, 35) |
| | 90th percentile | 218 (ref) | **−71** (−102, −41) | **−60** (−85, −34) | **−57** (−66, −48) | **−41** (−66, −16) | **79** (66, 93) | 8 (−15, 30) | **−17** (−34, −1) | −21 (−67, 26) | **−39** (−67, −10) |

Continued

## Table 6 Continued

| Total interval (all) | | Wales | England | Scotland | Northern Ireland | Denmark | Ontario | Norway | Manitoba | Victoria | Sweden |
|---|---|---|---|---|---|---|---|---|---|---|---|
| | n | 217 | 343 | 337 | 291 | 320 | 354 | 298 | 277 | 327 | 253 |
| | Median | 60 (ref) | **−9** (−14, −3) | −4 (−11, 2) | **−13** (−18, −8) | **−15** (−26, −4) | **16** (6, 26) | **−13** (−21, −6) | **17** (9, 24) | **−18** (−24, −12) | **−17** (−25, −10) |
| | 75th percentile | 81 (ref) | **−12** (−20, −4) | 1 (−9, 10) | **−13** (−20, −6) | **−14** (−20, −9) | **32** (24, 41) | −4 (−12, 4) | **35** (21, 48) | **−19** (−25, −12) | **−17** (−26, −7) |
| | 90th percentile | 123 (ref) | −13 (−34, 9) | 3 (−12, 18) | −5 (−22, 11) | −8 (−38, 22) | **73** (33, 114) | **32** (2, 62) | **47** (13, 82) | −12 (−37, 13) | −24 (−51, 4) |

The differences for the median, 75th and 90th percentiles are calculated as marginal effects after quantile regression by setting the continuous covariate age to its mean value and comorbidity to the mode. Significant results are shown in bold. Note that analyses of primary care intervals are presented in the table. The actual number of days included for Wales is shown in table 5.

ref, reference.

The overall response rate of 40% was comparable with similar studies among patients and PCPs,[30] but varied markedly between jurisdictions, with a response rate of 78% in Denmark and 24% in Norway. This may have resulted in differential selection bias across jurisdictions. However, comparing the participants on a number of variables (eg, tumour stage, presenting symptom, comorbidity, self-assessed health, smoking status) did not show any meaningful differences. Nevertheless, as recruitment strategies differed, with some women contacted directly by the registries and others via PCPs (and via nurses in Northern Ireland) assessing eligibility, some selection bias might have been introduced through differing impact of self-selection. This is underlined by the higher-than-expected proportion of stage I tumours among our respondents, when compared with an earlier registry-based breast cancer study.[3] Women with early-stage breast cancer are more able to participate in studies of this kind, compared with women with late-stage disease.

The statistical precision of the study was sufficient as we were able to show clinically significant differences of 1 week in time intervals, appropriate for clinically relevant differences.

### Comparison with other studies

Median patient intervals for breast cancer have varied in previous studies; in Denmark, the interval was 14 days in a 2004 study[30] and only 3 days in 2015 (using similar methods),[31] possibly reflecting cancer pathway improvements over this period. Other studies have identified the interval as 13 days (UK),[32] 14 days (New Zealand),[33] 16 days (Germany)[34] and 7 days (Sweden).[35] Reasons for discrepancies may reflect a difference in the time-points and data sources used to define patient interval or changes over time in access to primary care.

Diagnostic intervals vary; a UK study on patients diagnosed in 1999–2000 found a median diagnostic interval of 30 days,[36] while other UK studies found a diagnostic interval of 14–27 days.[37–39] Lower intervals in more recent studies possibly reflect the introduction of urgent referrals in 2010. A Danish study demonstrated an 18-day interval in 2010,[39] compared with 8 days in our study.[39] Similarly, a study in Manitoba in 2009–2010 demonstrated an interval of 35 days compared with 28 days in our study.[40] The primary care interval has had less attention in the literature, although a Manitoban study reported an interval of 15 days, comparable with our finding of 17 days.[40] It also reported a total interval of 78 days, which is comparable with the 76 days found in our study. A recent Swedish study focusing on the time from referral to treatment, which roughly corresponds to a combined diagnostic and treatment interval, reported a median interval of 20 days, compared with 35 days in our study.[35]

Clinical referral for a suspected breast cancer (eg, lump) is often expedited.[37 38] Thus, the high proportion of women with 'red flag' symptoms or signs may mean that time intervals for breast cancer diagnosis vary less than other common cancers, as differential diagnosis

is easier and investigations and referral pathways more straightforward. Nevertheless, we found important variations in intervals for symptomatic women, possibly reflecting methodological differences, but also temporal and international differences in diagnostic pathways. We would expect these differences to primarily impact on the median primary care interval, 0 days for all jurisdictions except Manitoba, Ontario and Victoria. Note, however, that in these jurisdictions primary care is more often responsible for confirming the breast cancer diagnosis, thus extending the interval. Indeed, a study from Manitoba confirms that PCPs experience significant waiting time when they order a mammogram.[40]

## CONCLUSION AND IMPLICATIONS FOR CLINICAL PRACTICE AND HEALTHCARE ORGANISATION

Despite the existence of well-established screening programmes, the diagnostic route for breast cancer remains critically important, as over half of patients are diagnosed this way. Awareness and recognition of warning signs impact patient intervals[41]; we found, in some jurisdictions, significant numbers of women waiting more than 90 days before presenting with symptoms. Targeted awareness interventions could facilitate more timely diagnosis of breast cancer. The variation in waiting times for an appointment in primary care, which might affect perceived access, could also be addressed.

The possible impact of variations in routes to diagnosis, diagnostic/treatment intervals and key outcomes (eg, disease-free survival and mortality) cannot be assessed from our study. There is, nevertheless, growing evidence of an association between time intervals, the use of urgent referrals and mortality.[10 41–44] More research are warranted to explore this relationship.

This study illustrates the need to optimise diagnostic routes for breast cancer internationally. Awareness of international differences in key time intervals is an important step in optimising pathways. Ideally, routine and standardised collection of time interval and route to diagnostic information will become the norm internationally; this will greatly assist in optimisation of breast cancer diagnostic pathways.

**Author affiliations**
[1]Department for Clinical Medicine, Aarhus Universitet, Aarhus, Denmark
[2]Department of Public Health, Research Unit for General Practice, Aarhus University, Aarhus C, Denmark
[3]General Practice, University of Edinburgh, Edinburgh, UK
[4]Gynaecological Cancer Research Centre, University College London, London, UK
[5]Scottish Registry, Information Services Division, NHS National Services Scotland, Edinburgh, UK
[6]Department of Epidemiology and Health Statistics, School of Public Health, Fujian Medical University, Fuzhou, Fujian, China
[7]N Ireland Cancer Registry, Queen's University Belfast, Belfast, UK
[8]Renal Network, Cancer Care Ontario, Toronto, Ontario, Canada
[9]Department of Family and Community Medicine, Ontario Institute for Cancer Research, Toronto, Ontario, Canada
[10]University Hospital, Regional Cancer Centre of Central Sweden, Uppsala, Sweden
[11]Department of Oncology, Lund University Hospital, Lund, Sweden
[12]Population Oncology, Cancer Care Manitoba, Winnipeg, Manitoba, Canada
[13]Department of Epidemiology and Cancer Registry, CancerCare Manitoba, Winnipeg, Manitoba, Canada
[14]Patient Navigation, CancerCare Manitoba, Winnipeg, Manitoba, Canada
[15]North Wales Centre for Primary Care Research, Bangor University, Bangor, UK
[16]University of Leeds, Leeds, UK
[17]CBRC, Cancer Council Victoria, Melbourne, Victoria, Australia
[18]Deakin University Faculty of Health, Burwood, Victoria, Australia
[19]Centre for Behavioural Research in Cancer, Cancer Council Victoria, Melbourne, Victoria, Australia
[20]Policy and Information, Cancer Research UK, London, UK
[21]Women's Cancer, University College London, London, UK

**Acknowledgements** We thank our colleagues and the participating patients, primary care physicians and cancer treatment specialists in all participating ICBP jurisdictions; Catherine Foot, Martine Bomb, Brad Groves, Irene Reguilon, Charlotte Lynch and Samantha Harrison of Cancer Research UK for managing the programme; John Butler, Royal Marsden NHS Foundation Trust and ICBP Clinical Lead, for his advice; and the ICBP Module 4 Academic Reference Group for providing independent peer review and advice for the study protocol and analysis plan development. We would like to thank the staff at the Information Services Division of NHS National Services Scotland for identifying potential study participants in Scotland and mailing patient questionnaires to primary care physicians, as well as the Danish Breast Cancer Group (DBCG) for providing secondary care (specialist) data to this study.

**Collaborators** *ICBP Module 4 Working Group:* Alina Zalounina Falborg (Statistician, Research Unit for General Practice, Department of Public Health, Aarhus University, Aarhus C, Denmark); Andriana Barisic (Research Associate, Department of Prevention and Cancer Control, Cancer Care Ontario, Toronto, Ontario, Canada); Anna Gavin (Director, Northern Ireland Cancer Registry, Centre for Public Health, Queen's University Belfast, Belfast, UK); Anne Kari Knudsen (Administrative Leader, Department of Cancer Research and Molecular Medicine, Norwegian University of Science and Technology, Trondheim, Norway); Breann Hawryluk (Project Planning Coordinator, Department of Patient Navigation, CancerCare Manitoba, Winnipeg, Manitoba, Canada); Chantelle Anandan (Research Fellow, Centre for Population Health Sciences, University of Edinburgh, Edinburgh, UK); Conan Donnelly (Research Fellow, Centre for Public Health, Queen's University Belfast, Belfast, UK); David Brewster (Scottish Cancer Registry, Information Services Division, NHS National Services Scotland, Edinburgh, UK); David Weller (James Mackenzie Professor of General Practice, Centre for Population Health Sciences, University of Edinburgh, Edinburgh, UK); Donna Turner (Epidemiologist/Provincial Director, Population Oncology, CancerCare Manitoba, Winnipeg, Manitoba, Canada); Elizabeth Harland (Project Coordinator, Department of Epidemiology and Cancer Registry, CancerCare Manitoba, Winnipeg, Manitoba, Canada); Eva Grunfeld (Director, Knowledge Translation Research Network Health Services Research Program, Ontario Institute for Cancer Research; Professor and Vice Chair of Research, Department of Family and Community Medicine, University of Toronto, Toronto, Ontario, Canada); Evangelia Ourania Fourkala (Research Associate, Gynaecological Cancer Research Centre, Women's Cancer, Institute for Women's Health, University College London, UK); Henry Jense (Research Fellow, Research Unit for General Practice, Department of Public Health, Aarhus University, Aarhus C, Denmark); Jackie Boylan (Research Fellow, Centre for Public Health, Queen's University Belfast, Belfast, UK); Jacqueline Kelly (Tumour Verification Officer, Northern Ireland Cancer Registry, Centre for Public Health, Queen's University Belfast, Belfast, UK); Kerry Moore (Research Fellow, Centre for Public Health, Queen's University Belfast, Belfast, UK); Maria Rejmyr Davis (Head, Southern Sweden Regional Cancer Center, Lund, Sweden); Martin Malmberg (MD PhD, Senior Consultant, Department of Oncology, Lund University Hospital, Lund, Sweden); Mats Lambe (Professor of Medical Epidemiology, Regional Cancer Center Uppsala and Department of Medical Epidemiology and Biostatistics, Karolinska Institutet, Stockholm, Sweden); Oliver Bucher (Epidemiologist, Department of Epidemiology and Cancer Registry, CancerCare Manitoba, Winnipeg, Manitoba, Canada); Peter Vedsted (Professor, Research Unit for General Practice, Department of Public Health, Aarhus University, Aarhus C, Denmark); Rebecca Bergin (Senior Research Officer/PhD Candidate, Centre for Behavioural Research in Cancer, Melbourne, Victoria, Australia); Rebecca-Jane Law (Research Project Support Officer, North Wales Centre for Primary Care Research, Bangor University, Wrexham, UK); Richard D Neal (Professor of Primary Care Oncology, Academic Unit of Primary Care, Leeds Institute of Health Sciences, University of Leeds, Leeds, UK); Sigrun Saur Almberg (Researcher, Department of Cancer Research and Molecular Medicine, Faculty of Medicine, Norwegian University of Science and Technology

(NTNU), Trondheim, Norway); Therese Kearney (Research Fellow, Northern Ireland Cancer Registry, Centre for Public Health, Queen's University Belfast, Belfast, UK); Jatinderpal Kalsi (Project Manager, Gynaecological Cancer Research Centre, Women's Cancer, Institute for Women's Health, University College London, UK); Victoria Cairnduff (Statistician, Northern Ireland Cancer Registry, Centre for Public Health, Queen's University Belfast, Belfast, UK); Victoria Hammersley (Researcher, Centre for Population Health Sciences, University of Edinburgh, Edinburgh, UK); Victoria White (Deputy Director, Centre for Behavioural Research in Cancer, Cancer Council Victoria, Melbourne, Victoria, Australia); Usha Menon (Professor of Gynaecological Oncology and Head, Gynaecological Cancer Research Centre, Women's Cancer, Institute for Women's Health, University College London, UK); Yulan Lin (Postdoc, Department of Cancer Research and Molecular Medicine, Faculty of Medicine, Norwegian University of Science and Technology (NTNU), Trondheim, Norway).*ICBP Module 4 Academic Reference Group:* Professor Jan Willem Coebergh (Professor of Cancer Surveillance, Department of Public Health, Erasmus Universiteit Rotterdam, Rotterdam, The Netherlands); Jon Emery (Professor of Primary Care Cancer Research, University of Melbourne; Clinical Professor of General Practice, University of Western Australia, Australia); Dr Stefan Bergström (Senior Consultant Oncologist, Department of Oncology, Gävle, Sweden); Dr Monique E van Leerdam (Erasmus MC University Medical Center, The Netherlands); Professor Marie-Louise Essink-Bot (Academic Medical Centre, Amsterdam University, The Netherlands); Professor Una MacLeod (Senior Lecturer in General Practice and Primary Care, Hull York Medical School, UK).

**Contributors** UM, HJ, PV, DW: planned the study design, data collection, carried out the analyses and wrote the draft manuscript, and were also responsible for local data collection (alongside the Working Group), management and interpretation, and participated in writing and approval of the final manuscript. AZF: planned the study design, data collection, carried out the analyses and wrote the draft manuscript. JK, DB, YL, AG, AB, EG, ML, MM, DT, EH, BH, R-JL, RDN, VW, RB, SH: responsible for local data collection (alongside the ICBP Module 4 Working Group), management and interpretation, and participated in writing and approval of the final manuscript. ICBP Module 4 Working Group: responsible for local data collection. PV, DW: responsible for the overall content as the guarantor.

**Funding** Funding for ICBP Module 4 was provided by CancerCare Manitoba, Cancer Care Ontario, Canadian Partnership Against Cancer (CPAC), Cancer Council Victoria, Cancer Research Wales, Cancer Research UK, Danish Cancer Society, Danish Health and Medicines Authority, European Palliative Care Research Centre (PRC), Norwegian University of Science and Technology (NTNU), Guidelines and Audit Implementation Network (GAIN), Macmillan Cancer Support, National Cancer Action Team, NHS England, Northern Ireland Cancer Registry funded by the Public Health Agency NI, Norwegian Directorate of Health, Research Centre for Cancer Diagnosis in Primary Care (CaP) at Aarhus University in Denmark, Scottish Government, Swedish Association of Local Authorities and Regions, University College London and NIHR Biomedical Research Centre at University College London NHS Foundation Trust, University of Edinburgh (R42856), Victorian Department of Health and Human Services, and Welsh Government. The funding bodies had no influence on the design of the study, on the collection, analysis and interpretation of data, on writing the manuscript, or whether to publish the results.

**Competing interests** None declared.

**Patient and public involvement** Patients and cancer charities participated in the development of the idea and the questionnaire and the data collection. The researchers were involved in the design, analyses and writing of this research.

**Patient consent for publication** Not required.

**Ethics approval** This study involves human participants and was approved by the Cancer Council Victoria Human Research Ethics Committee (HREC 1125); Health Research Ethics Board, University of Manitoba Research Resource Ethics Committee; CancerCare Manitoba (HS15227) (H2012:105)); RRIC#28-2012; University of Toronto Research Ethics Board (27881); and Danish Data Protection Agency. According to Danish law and the Central Denmark Region Committees on Health Research Ethics, approval by the National Committee on Health Research Ethics was not required as no biomedical intervention was performed (2013-41-20301-10-72-20-13). The study was also approved by the Ethics Review Board, Uppsala (2013/306); Regional Committees for Medical and Health Research Ethics (2013/136/REK nord); NRES Committee East Midlands - Derby 2, local R&D for each health board (11/EM/0420); and Privacy Advisory Committee, CHI Advisory Group (11/EM/0420ORECNI). Ethical approval was also obtained from the local governance of each health trust (12/NI/0053) and the NRES Committee East Midlands - Derby 2 R&D for each clinical research network (11/EM/0420). Participants gave informed consent to participate in the study before taking part.

**Provenance and peer review** Not commissioned; externally peer reviewed.

**Data availability statement** All data relevant to the study are included in the article or uploaded as supplementary information.

**ORCID iDs**
Henry Jensen http://orcid.org/0000-0003-4040-7334
Rebecca-Jane Law http://orcid.org/0000-0002-1435-5086
Richard D Neal http://orcid.org/0000-0002-3544-2744
Usha Menon http://orcid.org/0000-0003-3708-1732

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
