## [Reviewer comments · BMJ Open]

ARTICLE DETAILS

TITLE (PROVISIONAL)	Diagnostic pathways for breast cancer in ten International Cancer Benchmarking Partnership (ICBP) jurisdictions: An International comparative cohort study based on questionnaire and registry data
AUTHORS	Vedsted, Peter; Weller, David; Zalounina Falborg, Alina; Jensen, Henry; Kalsi, Jatinderpal; Brewster, David; Lin, Yulan; Gavin, Anna; Barisic, Andriana; Grunfeld, Eva; Lambe, Mats; Malmberg, Martin; Turner, Donna; Harland, Elizabeth; Hawryluk, Breann; Law, Rebecca-Jane; Neal, Richard; White, Victoria; Bergin, Rebecca; Harrison, Samantha; Menon, Usha; Working Group, ICBP

VERSION 1 – REVIEW

REVIEWER	Pal Choudhury, Parichoy National Cancer Institute Division of Cancer Epidemiology and Genetics, Trans-divisional Research Program
REVIEW RETURNED	29-Dec-2021

GENERAL COMMENTS	The authors have presented a comprehensive and well conducted analysis evaluating differences in diagnostic pathways in terms of patient interval, primary care interval, diagnostic interval and treatment interval, across 10 regions in 6 countries. I just have one comment on the quantile regression approach used to estimate the differences in interval lengths between Wales (reference) and the other regions. 1. Generally, I think it will benefit the readers if the authors include some details of the quantile regression model used, like the mathematical equation, at least as a supplement. From the tables, I figured that the covariates included were age and comorbidities. I wondered if the authors considered adjusting for additional covariates in Table 1. Also looking at estimates and confidence intervals of the regression coefficients could be informative about which covariates are driving the differences.
--

REVIEWER	Dodwell, David Oxford University, Nuffield Department of Population Health
REVIEW RETURNED	05-Jan-2022

GENERAL COMMENTS	The study represents a colossal research effort but there are several issues to address Major concerns There are several strategic concerns concerning this study which are acknowledged by the study authors. The major one in my view is
---

	that respondents to the questionnaires may well be unrepresentative of the generality of patients who develop breast cancer and therefore the magnitude of the components of delay between first symptom (or screening) and start of treatment. Patients with a greater symptom burden and/or requiring more intensive treatment (ie those with more advanced disease) are less likely to complete questionnaires. This makes it difficult to identify an overarching strategy to reduce delays, particularly as the response rates differ so much between jurisdictions. The relevance to the totality of patients who develop breast cancer is therefore limited. Supplementary table 1 is unintelligible (at least to me!) and a consort diagram would be better to help aid understanding of the study design and response rates. Within the discussion the advice on interventions to reduce delay is very non-specific and of little practical value. This is understandable given the different health care systems involved but does raise a question over the practical utility of this research. Minor concerns The content and access to the 'Additional files' is unclear. Are these to be made available to readers and if so how? They have not been included in the files to be peer reviewed. There are several syntactical errors, and the manuscript requires careful re-reading to identify and correct these. All 'symptomatic' patients should have a symptom! (page 10, line 264) The choice of Wales as the reference for Figure 2 is not justified. It's not clear why recall bias is reduced by triangulation of data. (page 1, line 31) It's not clear why differences in patient survival affect patient experience (page 1, line 45)
--	---

VERSION 1 – AUTHOR RESPONSE

Reviewer: 1

The authors have presented a comprehensive and well conducted analysis evaluating differences in diagnostic pathways in terms of patient interval, primary care interval, diagnostic interval and treatment interval, across 10 regions in 6 countries. I just have one comment on the quantile regression approach used to estimate the differences in interval lengths between Wales (reference) and the other regions.

Generally, I think it will benefit the readers if the authors include some details of the quantile

regression model used, like the mathematical equation, at least as a supplement. From the tables, I figured that the covariates included were age and comorbidities. I wondered if the authors considered adjusting for additional covariates in Table 1. Also looking at estimates and confidence intervals of the regression coefficients could be informative about which covariates are driving the differences.

RESPONSE: Thank you for this suggestion to include more details on the regression used. We have added a short introduction to quantile regression for count data (Supplementary Box 1). We considered adjusting for additional covariates but opted against this due to some of the covariates not being defined in the same way across jurisdictions – they could not, therefore, be used in a common model. Additionally, due to the number of cases we opted for the least possible covariates to avoid small cell counts in the model - a prerequisite of the model. Regarding showing estimates of the covariates - in the model used (quantile regression with marginal means), the estimates of the covariates cannot be used to interpret what drives the effect, as all covariates are set at their mode (most common value), and the quantile regression uses weights of these to calculate the estimates of the variable of interest (time interval). This means the estimates of the covariates merely depict an "average" contribution across time and jurisdictions. Therefore, showing the estimates for the covariates normally is appreciated, but doing so in a quantile regression would most likely lead to misinterpretation.

Note that we now refer to covariate issue in the Discussion section of the paper.

Reviewer: 2

The study represents a colossal research effort but there are several issues to address.

There are several strategic concerns concerning this study which are acknowledged by the study authors. The major one in my view is that respondents to the questionnaires may well be unrepresentative of the generality of patients who develop breast cancer and therefore the magnitude of the components of delay between first symptom (or screening) and start of treatment. Patients with a greater symptom burden and/or requiring more intensive treatment (ie those with more advanced disease) are less likely to complete questionnaires. This makes it difficult to identify an overarching strategy to reduce delays, particularly as the response rates differ so much between jurisdictions. The relevance to the totality of patients who develop breast cancer is therefore limited.

REPSONSE: We appreciate those concerns. We have recognised this potential bias towards patients with early stage disease being more likely to participate in this type of study in the Strengths & Weaknesses section in the manuscript. We have included some additional text examining the implications of the potential bias. Future studies may be less dependent on response rates to surveys such as ours – eg they may be able to use more routinely collected data (which we now advocate in the paper).

Nevertheless, regardless of selection bias, the existence of delays for any patient population is an important consideration which needs to be better understood and addressed. This study helps identify the countries where patients have experienced longer delays and where the potential driver of delay exists along the patient pathway – this is an important exercise to support healthcare systems across different countries in identifying how to develop strategies to reduce delays. Initiatives for early stage patients may also be relevant for advanced stage patients. This study provides the baseline data and insight to enable countries to reflect, improve and optimise their pathways for breast cancer patients.

Supplementary table 1 is unintelligible (at least to me!) and a consort diagram would be better to help aid understanding of the study design and response rates.

RESPONSE: We agree that a description of patient flow can be done in a more understandable way.

We have now drawn a CONSORT flow diagram instead, thank you for pointing this out (Supplementary Figure 1).

Within the discussion the advice on interventions to reduce delay is very non-specific and of little practical value. This is understandable given the different health care systems involved but does raise a question over the practical utility of this research.

RESPONSE: The primary aim of this work is to benchmark time intervals and routes to diagnosis to identify where certain countries may need to address the development of strategies to improve more timely diagnosis. The data generated will help inform those countries where this is required. We have highlighted broad advice on interventions, as you have identified, as analysing the specific needs for improvements and interventions in each country is a substantial piece of work, requiring its own manuscript. Nevertheless, in response to your feedback, we have enhanced this section. Importantly, the ICBP continues to support the countries involved following publication of all studies by facilitating knowledge transfer and lessons learnt in areas of need – this applies to this paper which will serve as a baseline to identify where further collaboration is needed to drive improvements in service.

The content and access to the 'Additional files' is unclear. Are these to be made available to readers and if so how? They have not been included in the files to be peer reviewed.

RESPONSE: Apologies for this, there may have been an error during the submission process. These are now included and will be available as part of the manuscript.

There are several syntactical errors, and the manuscript requires careful re-reading to identify and correct these.

RESPONSE: Apologies for this, we have now put the manuscript through several further rounds of review to check this and rectify any issues.

All 'symptomatic' patients should have a symptom! (page 10, line 264)RESPONSE:

Apologies, this was an error and has been rectified.

The choice of Wales as the reference for Figure 2 is not justified

RESPONSE: Wales was chosen as the reference jurisdiction as it had the lowest ovarian cancer survival in the ICBP Module 1 cancer survival analysis. This explanation has now been included in the footnote for Figure 2

It's not clear why recall bias is reduced by triangulation of data. (page 1, line 31)

RESPONSE: The authors agree this could use further explanation. We have now restructured this sentence and added some further clarity to rectify this.

It's not clear why differences in patient survival affect patient experience (page 1, line 45)

RESPONSE: We agree. We have rephrased this to include 'patient satisfaction with care' as the references listed indicate. Apologies for any confusion.

We hope this revised manuscript meets the reviewers' concerns, and that you now consider the paper suitable for publication. We would be happy, of course, to respond to any further queries

VERSION 2 – REVIEW

REVIEWER	Dodwell, David Oxford University, Nuffield Department of Population Health
REVIEW RETURNED	09-Apr-2022
GENERAL COMMENTS	Despite previous review the manuscript remains difficult to read because of many problems with syntax and grammar. The authors for whom English is their first language need to spend time rechecking the text carefully. There are two additional files are incorrectly labelled. Clearly a colossal amount of work has gone into this study and it is important and the results should be made available to inform future research and efforts to address delays in referral and diagnosis of breast cancer, but further effort is needed to improve the presentation and readability of this work.

VERSION 2 – AUTHOR RESPONSE

Thank you very much for the helpful suggestion to review syntax/gramma. We have given the paper a thorough scientific edit and have received approval from our co-authors.